# The Cloning and Characterization of a Three-Finger Toxin Homolog (NXH8) from the Coralsnake *Micrurus corallinus* That Interacts with Skeletal Muscle Nicotinic Acetylcholine Receptors

**DOI:** 10.3390/toxins16040164

**Published:** 2024-03-22

**Authors:** Henrique Roman-Ramos, Álvaro R. B. Prieto-da-Silva, Humberto Dellê, Rafael S. Floriano, Lourdes Dias, Stephen Hyslop, Raphael Schezaro-Ramos, Denis Servent, Gilles Mourier, Jéssica Lopes de Oliveira, Douglas Edgard Lemes, Letícia V. Costa-Lotufo, Jane S. Oliveira, Milene Cristina Menezes, Regina P. Markus, Paulo Lee Ho

**Affiliations:** 1Laboratório de Biotecnologia, Programa de Pós-Graduação em Medicina, Universidade Nove de Julho (UNINOVE), São Paulo 01504-001, SP, Brazil; hdelle@uni9.pro.br (H.D.); jessica.lo@uni9.edu.br (J.L.d.O.); douglas.edgard@uni9.edu.br (D.E.L.); 2Laboratório de Genética, Instituto Butantan, São Paulo 05503-900, SP, Brazil; alvaro.prieto@butantan.gov.br; 3Laboratório de Toxinologia e Estudos Cardiovasculares, Universidade do Oeste Paulista (UNOESTE), Presidente Prudente 19067-175, SP, Brazil; rafael@unoeste.br; 4Departamento de Farmacologia, Faculdade de Ciências Médicas, Universidade Estadual de Campinas (UNICAMP), Campinas 13083-887, SP, Brazil; phdlourdes@gmail.com (L.D.); hyslop@unicamp.br (S.H.); raphaelschezaroramos@gmail.com (R.S.-R.); 5Service d’Ingénierie Moléculaire pour la Santé (SIMoS), Département Médicaments et Technologies pour la Santé, Université Paris Saclay, Commissariat à l’énergie Atomique et aux Énergies Alternatives (CEA), F-91191 Gif sur Yvette, France; denis.servent@cea.fr (D.S.); mourier.g@gmail.com (G.M.); 6Departamento de Farmacologia, Instituto de Ciências Biomédicas, Universidade de São Paulo (USP), São Paulo 05508-000, SP, Brazil; costalotufo@gmail.com; 7Centro de Biotecnologia, Instituto Butantan, São Paulo 05503-900, SP, Brazil; janesoliveira@usp.br; 8Centro Bioindustrial, Instituto Butantan, São Paulo 05503-900, SP, Brazil; milene.santos@butantan.gov.br; 9Laboratório de Cronofarmacologia, Instituto de Biociências, Universidade de São Paulo (USP), São Paulo 05508-090, SP, Brazil; rpmarkus@usp.br

**Keywords:** *Micrurus corallinus*, coralsnake, three-finger toxins, neuromuscular blockade, nAChR

## Abstract

Coralsnakes (*Micrurus* spp.) are the only elapids found throughout the Americas. They are recognized for their highly neurotoxic venom, which is comprised of a wide variety of toxins, including the stable, low-mass toxins known as three-finger toxins (3FTx). Due to difficulties in venom extraction and availability, research on coralsnake venoms is still very limited when compared to that of other Elapidae snakes like cobras, kraits, and mambas. In this study, two previously described 3FTx from the venom of *M. corallinus*, NXH1 (3SOC1_MICCO), and NXH8 (3NO48_MICCO) were characterized. Using in silico, in vitro, and ex vivo experiments, the biological activities of these toxins were predicted and evaluated. The results showed that only NXH8 was capable of binding to skeletal muscle cells and modulating the activity of nAChRs in nerve–diaphragm preparations. These effects were antagonized by anti-rNXH8 or antielapidic sera. Sequence analysis revealed that the NXH1 toxin possesses eight cysteine residues and four disulfide bonds, while the NXH8 toxin has a primary structure similar to that of non-conventional 3FTx, with an additional disulfide bond on the first loop. These findings add more information related to the structural diversity present within the 3FTx class, while expanding our understanding of the mechanisms of the toxicity of this coralsnake venom and opening new perspectives for developing more effective therapeutic interventions.

## 1. Introduction

According to the World Health Organization (WHO), snakebites are a neglected public health concern in tropical and subtropical regions of the world [1], with envenomation by snakes of the families Viperidae and Elapidae being the most important because of the frequently severe clinical manifestations [2].

With over 70 species distributed throughout the southeastern United States and South America, coralsnakes (*Micrurus* spp.), which are known for their distinctive coloration of red, white, and black rings, are the only terrestrial elapids found in the New World [3,4]. However, the low aggressivity of these snakes, their fossorial habits, and their difficulty in inoculating venom means that coralsnake bites are relatively uncommon [5]. Thus, of the >20,000 snakebites per year in Brazil, <1% are caused by coralsnakes [5,6]. Neurotoxic symptoms such as muscle weakness, ophthalmoplegia, diplopia, and palpebral ptosis are among the most common manifestations in systemic envenomation [4,5,6].

Coralsnake venoms are highly neurotoxic [6,7,8,9,10] and consist of a combination of toxins that primarily cause neuromuscular blockade, but can also elicit other physiological effects, such as edema [11], hemolysis [11,12], myotoxicity [7,13,14], cardiotoxicity [15,16], and nephrotoxicity [14]. Given the potential severity of the neurotoxic manifestations, including the possibility of respiratory failure leading to death [5,6,17], antivenom therapy is not only the most recommended and effective treatment for neutralizing the coralsnake venom, but is also deemed an essential drug by a WHO expert committee [18]. 

Despite the crucial role of antivenom therapy, shortages of coralsnake antivenom are frequently reported [17,19,20]. These shortages are partly attributable to the generally small size of most *Micrurus* spp. and their low venom yields [21], and the difficulties in capturing and maintaining these snakes in captivity for venom production [3,22]. In Brazil, the bulk of venom collected for antivenom production by producers such as the Instituto Butantan (São Paulo, SP, Brazil) and Fundação Ezequiel Dias (FUNED) (Belo Horizonte, MG, Brazil) is obtained from two species—*M. corallinus* and *M. frontalis*—that account for most of the reported coralsnake bites in this country [5,23]. Consequently, the amount of venom available for research is minimal.

In spite of the aforementioned challenges, recent studies have considerably advanced our understanding of the composition of *Micrurus* venoms. Transcriptomic [24,25], proteomic [25,26,27], and biochemical [28] studies have shown that these venoms consist of two main groups of toxins: (i) phospholipase A_2_ (PLA_2_), including β-neurotoxins that cause neuromuscular blockade by inhibiting the presynaptic release of acetylcholine (ACh) from nerve terminals, and (ii) three-finger toxins (3FTx): a diverse group of small, non-enzymatic toxins widely distributed in the families Elapidae and Hydrophiidae (sea snakes) [29,30,31].

With a primary structure rich in cysteine residues, all 3FTx contain different conserved residues (Gly^22^, Tyr/Phe^27^, Gly^47^, Pro^53^, and Asn^71^), and at least four conserved disulfide bonds (Cys^3^/Cys^26^, Cys^19^/Cys^48^, Cys^52^/Cys^64^, and Cys^65^/Cys^70^) that are essential for the characteristic stable scaffold formed by the three major loops. Despite the shared tertiary structure of three loops that form a flat, triple-stranded anti-parallel β sheet [32], the 3FTx family consists of toxins with a wide range of biological activities [33,34,35,36,37], including acetylcholinesterase inhibitors (also known as *fasciculins*) [38], curaremimetic α-neurotoxins [34,37,39,40], cardiotoxins [15,41], cytotoxins [42,43], muscarinic toxins [44,45], platelet aggregation inhibitors [46], sodium- [47] and potassium [48]-channel activators, as well as ASIC [49] and L-type calcium-channels blockers [50].

Among the curaremimetic 3FTx, long-chain α-neurotoxins are characterized by an additional S-S bond between Cys^32^ and Cys^36^ in the second loop, and are particularly noteworthy for their potent inhibition of αβγδ nAChRs [34,39], while also exhibiting the capability of binding to neuronal α7 nAChRs [51]. In contrast, κ-Neurotoxins, such as κ-BgTx from *Bungarus multicinctus* [52], are a subclass of dimeric 3FTx that differs from other postsynaptic α-neurotoxins in their specific inhibitory action on neuronal α3 and α4 nAChRs, while lacking the ability to bind to muscle nAChRs [52,53].

An additional category of 3FTx found only in the venoms of elapids such as *Naja* [54,55,56,57,58,59] and *Bungarus* [60,61,62,63,64,65,66] species consists of non-conventional toxins (also known as miscellaneous-type toxins, melanoleuca-type toxins, or long α-neurotoxin-like homologs) that have ten cysteine residues and a five-disulfide bridge structure, but, unlike the long α-neurotoxins and κ-neurotoxins, the fifth disulfide bond is located between Cys^6^ and Cys^11^ in the N-terminal of the first loop.

While non-conventional toxins are also often referred to as “weak toxins” because of their lower toxicity in mice (LD_50_ of 5–80 mg/kg, i.v.) [57], their actual toxicity can vary greatly, and, in some cases, it may be comparable to that of α-neurotoxins, e.g., γ-bungarotoxin (LD_50_ of 0.15 mg/kg, i.v.) [63]. Moreover, although their mechanisms of action are not yet fully understood, these toxins may interact with different receptors and ion channels in the body, potentially affecting nerve and muscle functions, blood pressure, and coagulation.

Compared to the extensive knowledge available for members of the 3FTx family in elapids in general, considerably less is known of the 3FTx of *Micrurus* spp., primarily because of the difficulty in obtaining venom for toxin purification. As of October 2023, a PubMed search using terms such as “*Micrurus* three-finger toxin” yielded only 42 studies, many of which focused on in silico analyses of transcriptomic and proteomic data, with few actually dealing with the biological and pharmacological characterization of these toxins; nevertheless, some reports have described the partial or complete characterization of short α-neurotoxins and phospholipases A_2_ [7,67,68,69,70,71,72,73].

Over the past two decades, we have investigated the venom composition of the Brazilian coralsnake *M. corallinus*. Two major groups of toxins are present in the venom: phospholipase A_2_ (PLA_2_) and three-finger toxins (3FTx). We have identified specific 3FTx toxins, such as NXH8, which is unique in its structure as it has an additional disulfide bond compared to other 3FTx, and NXH1, which has a similar structure to most other 3FTx [24,74,75,76]. The PLA_2_ toxin, on the other hand, is likely to be responsible for the venom’s pre-synaptic neurotoxicity, as previously described [9,10,30].

Using a combination of in silico and in vitro techniques, we have further identified potential immunogenic epitopes within the four major 3FTx from the venom of *M. corallinus*, including NXH8 and NXH7—which is related to NXH1 [74]—as well as potential reactive epitopes within the PLA_2_ [77]. These findings demonstrated the importance of these toxins for the overall toxicity of the venom and highlighted their potential as targets for the development of alternative and effective antivenom therapies.

Despite the reports indicated above, to date, no study has definitively demonstrated the potential function and targets of these toxins. Therefore, the present work’s aims are the following: (i) to conduct a sequence analysis of the NXH8 protein; (ii) to predict its biological activities through a cluster analysis based on the physicochemical properties of amino acid sequences of 3FTx with well-characterized biological activities; (iii) to use recombinant toxins (unfolded or partially refolded) expressed in *E. coli* as immunogens to produce polyclonal antibodies and evaluate their ability to prevent the binding of α-neurotoxins in the venom to muscle nAChR; (iv) to chemically synthesize NXH8 to confirm its mechanism of action on nAChR; and (v) to analyze the ability of antibodies generated against this neurotoxin to cross-react with other components of elapid venoms.

## 2. Results

### 2.1. cDNA and Protein Sequence Analysis

Venom gland transcripts from *M. corallinus* have revealed potential cDNAs encoding for various proteins of the 3FTx family [24,74,76]. The *nxh1* cDNA, while lacking critical amino acid residues for nAChR binding, exhibits considerable similarity to short-chain α-neurotoxins, with a conservation of 60–70% of residues [76]. Analysis of the *nxh8* cDNA sequence revealed a short segment (5 bp) at the 5′UTR (untranslated region), a 3′UTR containing a polyadenylation signal (204 bp), and an open reading frame (ORF; 258 bp) that encodes a protein characterized by a typical 21-residue signal peptide followed by a 65-residue sequence of the putative mature protein (Figure 1).

The phylogenetic clustering of the sequence alignments of NXH1 and NXH8 with several prototype toxin sequences from the 3FTx family yielded a dendrogram characteristic of the 3FTx family that consisted of at least 13 functional groups delineated by activity and 20 Orphan clades of as yet undetermined biological activity [36]. 

Despite sharing 42% similarity, NXH1 and NXH8 are distinct members of the 3FTx family in *M. corallinus* venom. NXH1 is a conventional 3FTx with the typical eight cysteine residues and is situated within the cluster known as Orphan group XII. In contrast, NXH8 features ten cystein residues, with a fifth disulfide bond presumably forming between Cys6 and Cys11. This structural feature confers NXH8, a pronounced similarity (~69%) with non-conventional toxins, such as those in *Bungarus* spp. venoms, e.g., candoxin from *B. candidus* and toxin BM10-1 from *B. multicinctus* [60,61,62] (Figure 2, Figure 3 and Figure 4), both classified in the *Bungarus* spp., Orphan group IV clade [36] (Figure 5). 

UniProt ID: LYNX1_MOUSE: Mature peptide from Lynx1, *Mus musculus*; 3NO4_BUNCA: Candoxin *B. candidus*; *B. multicinctus*; 3NO4H_BUNMU (*P15818*): Long neurotoxin homolog, *Bungarus multicinctus*; 3NO4H_BUNMU (*Q9YGI9*): Mature peptide Long neurotoxin homolog, *B. multicinctus*; 3NO48_MICCO: NXH8, *M. corallinus*; 3NO5I_BUNMU: γ-Bungarotoxin, *B. multicinctus*; 3NO56_BUNMU: Long neurotoxin homolog TA-bm16, *B. multicinctus*; 3NO26_NAJNA: Weak neurotoxin 6, *N. naja*; 3NO23_NAJAT: Weak neurotoxin NNAM3, *N. atra*; 3NO25_NAJNA: Weak neurotoxin 5, *Naja naja*; 3NO2B_NAJME: Toxin S4C11, *N. melanoleuca*; 3NO2B_NAJHH: Weak Toxin CM-11, *N. haje haje*; 3LKB_BUNMU: κ-Bungarotoxin, *B. multicinctus*; 3LKF_BUNFL: κ-Flavitoxin, *B. flaviceps flaviceps*; 3L21A_BUNMU: α-Bungarotoxin, *B. multicinctus*; 3L21_ASPSC: Long neurotoxin 1, *Aspidelaps scutatus*; 3L21_NAJKA: α-Cobratoxin, *N. naja kaouthia*; 3L21_NAJAC: Long neurotoxin III, *N. haje anchietae*; 3SIMA_DENPO: Muscarinic toxin-α, *D. polylepis polylepis*; 3SIM1_DENAN: Muscarinic toxin 1, *D. angusticeps*; 3SIM7_DENAN: Muscarinic toxin 7, *D. angusticeps*; 3SIY_DENJA: Toxin S2C4 chain 1, *D. jamesoni kaimosae*; 3SIY1_DENAN: Protein C8S2 Chain 1, *D. angusticeps*; 3SIYL_DENAN: Mature peptide from Synergistic-like venom protein, *D. angusticeps*; 3SOE_HEMHA: Cytotoxin homolog 9B, *Hemachatus haemachatus*; 3SA4_NAJAT: Cardiotoxin IV, *N. naja atra*; 3SA1_NAJMO: Cardiotoxin XIIB, *N. mossambica mossambica*; 3SE1_DENAN: Fasciculin-1, *D. angusticeps*; 3SEC_DENPO: Acetylcholinesterase toxin C, *D. polylepis polylepis*; 3SOC1_MICCO: NXH1, *M. corallinus*; 3S1EA_LATSE: Erabutoxin-a, *L. semifasciata*; 3S11_MICNI: Short neurotoxin alpha, *M. nigrocinctus*; 3S1CC_NAJAT: Cobrotoxin-b, *N. naja atra*; 3S11_NAJOX: Alpha neurotoxin, *N. oxiana*; 3SLS_DENPO: Calciseptine, *D. polylepis polylepis*; 3SL2_DENPO: Toxin FS-2, *D. polylepis polylepis*; 3SPM_DENJA: Mambin, *D. jamesoni kaimosae*; 3SP1_DENJA: Toxin S5C1, *D. jamesoni kaimosae*; 3NOJ_BUNCA: Bucandin, *B. candidus*; 3NOJ6_DENJA: Toxin S6C4, *D. jamesoni kaimosae*.

In agreement with this, the NXH8 toxin exhibits a 56.86% sequence similarity with weak neurotoxin 5 from *Naja naja*, and 50.70% similarity with WTX from *Naja kaouthia.* Both non-conventional toxins belong to the non-conventional weak neurotoxins’ Orphan group II clade [36] (Figure 2 and Figure 5).

NXH8 also shares a significant degree of amino acid sequence homology with short/long α-neurotoxins, including a 51.52% similarity to Erabutoxin-a, a short α-neurotoxin from *Laticauda semifasciata*, and a 49.35% similarity to Cobratoxin, a long α-neurotoxin from *Naja kaouthia*. These similarities, however, stand in contrast to the lower similarity that NXH8 has with Bucandin-type toxins (Orphan group XIX) [36], including Bucandin (33.33%) from *Bungarus candidus* and Toxin S6C6 (30.48%) from *Dendroaspis jamesoni kaimosae*, which, like NXH8, are non-conventional toxins featuring a fifth disulfide bond in the first loop.

The divergence of Bucandin-type toxins in the dendrogram, distant from NXH8, highlights an intriguing evolutionary pathway where structural novelties in these neurotoxins have emerged, possibly reflecting a broader functional adaptation in snake venomics. As a result, while NXH8 shares functional domains with traditional neurotoxins, the exact toxicological effects and biological role of this protein in *M. corallinus* envenomation remain to be elucidated.

### 2.2. Recombinant Expression of NXH8

While disulfide bonds are crucial for the stability and activity of 3FTx, obtaining biologically active toxins through recombinant expression in prokaryotic systems can be challenging, as the intracellular milieu of bacterial cells lacks the redox environment necessary for the formation of these chemical bonds.

In the present study, the recombinant expression of NXH8 was achieved by transforming *E. coli* BL21 (DE3) cells with the pRSETC-nxh8 plasmid (Figure 6a). As predicted, proteins were expressed as inclusion bodies that remained in the precipitate after the centrifugation of the cell lysate (Figure 6b).

The solubilization of the inclusion bodies resulted in high yields of purified rNXH8 under denaturing conditions (Figure 6b) or after purifying the protein following refolding (Figure 6c). Regardless of the purification method, a significant amount of protein aggregation occurred during dialysis (Figure 6d,e). 

When proteins were eluted through EDTA chelation, SDS-PAGE revealed a band of ~21 kDa (Figure 6c). Since the recombinant 6xHis-NXH8 fusion protein has a theoretical molecular mass of 11,399 Da, it is possible that this band may correspond to a dimer of rNXH8 [86]. Purified rNXH8 was inoculated into mice to generate polyclonal antibodies.

### 2.3. Cross-Reactivity of Anti-rNXH8 with Other Elapid and Non-Elapid Venoms

Polyclonal antibodies were generated in mice using a purified rNXH8 protein and were subsequently used in Western blotting to probe a large panel of *Micrurus* spp. and non-*Micrurus* spp. snake venoms (Figure 7). A robust reactive band was detected in the homologous venom of *M. corallinus* (lane 2, panel B), indicating the presence of NXH8 in this venom. An immunoreactive band was also seen in *M. altirostris* venom (lane 7) but was absent in the venoms of other *Micrurus* spp. This result indicates that polypeptides immunologically related to NXH8 are not widely distributed among *Micrurus* snakes.

The venoms of other elapids, such as *D. angusticeps* (eastern green mamba, lane 14) and *B. multicinctus* (many-banded krait, lane 16), also displayed immunoreactive bands. In contrast, the venom of *Notechis scutatus scutatus* (eastern tiger snake, lane 15), showed no immunoreactivity. A cardiotoxin from *N. n. kaouthia* venom (lane 18) that shares the general scaffold of 3FTx also did not cross-react with the anti-rNXH8 antibodies, a finding in agreement with the general absence of cardiotoxins in *Micrurus* venoms [30]. 

There was no cross-reactivity with two Viperidae venoms (*Crotalus durissus terrificus*—South American rattlesnake, lane 17, and *Bothrops jararaca*—jararaca, lane 19). Together, these findings indicate the presence of analogs immunologically related to NXH8 in various elapid venoms [76,87].

### 2.4. Components of M. corallinus Coralsnake Venom Bind to nAChR in Muscle-Membrane Preparations and Are Inhibited by Anti-NXH8 Antiserum

*Micrurus corallinus* venom interferes with neuromuscular transmission, primarily through the action of PLA_2_ β-neurotoxins that inhibit neurotransmitter release at the presynaptic membrane, and through postsynaptic α-neurotoxins that block acetylcholine binding to nAChR [9,10].

Figure 8 shows that incubation with *M. corallinus* venom markedly attenuated the binding of [^125^I]-labeled α-bungarotoxin to nAChR in rat neonatal skeletal muscle cell membranes, a finding indicative of the presence of toxins that compete with α-bungarotoxin for association with these receptors.

As indicated by cluster analysis, NXH8 could be a potential antagonist for nAChR. However, rNXH8 produced no toxic effects in the mice that were immunized to produce rNXH8 antiserum. This lack of toxicity does not mean that NXH8 is incapable of blocking nAChR but, rather, could reflect the expression of non-functional rNXH8 because of incorrectly folded toxins.

The preincubation of *M. corallinus* venom with polyclonal rNXH1 antibodies had no effect on the ability of the venom to interfere with the binding of [^125^I]-labeled α-bungarotoxin to neuromuscular nAChRs, suggesting that this antiserum had no neutralizing capacity against nAChR-binding toxins of the venom (Figure 8). In contrast, when *M. corallinus* venom was preincubated with polyclonal rNXH8 antibodies, the decrease in the binding of radio-labeled α-bungarotoxin to nAChRs caused by the two lowest venom concentrations was prevented, a finding strongly indicative of NXH8 being the main venom component that competes for binding to nAChR. 

However, antiserum to rNXH8 had no effect on the inhibitory action of the highest venom concentration tested (5 μg/mL), possibly because the volume of antiserum tested was insufficient to neutralize this amount of venom; the influence of a greater volume of antiserum was not assessed here.

The neutralizing effect of antiserum raised against a mixture (anti-MIX) of both toxins (rNXH1 and rNXH8) was similar to, but slightly less than, that observed with rNXH8 antibodies. Anti-MIX antibodies alone had no effect on the binding of radio-labeled α-bungarotoxin to nAChRs when compared to control binding (without antibodies) (Figure 8).

### 2.5. Chemically Synthesized NXH8 Causes Reversible Neuromuscular Blockades in Isolated Phrenic Nerve–Diaphragm Preparations

Mouse phrenic nerve–diaphragm (PND) preparations are routinely used to assess the neuromuscular activity of 3FTx, particularly that of postsynaptically active α-neurotoxins [88,89,90]. Figure 9f shows that synthetic NXH1 (sNXH1) exerted no blocking activity on nAChRs, a finding in agreement with the inability of polyclonal NXH1 antibodies to prevent the venom from interfering with the binding of radio-labeled a-bungarotoxin to these receptors (see previous section). This lack of activity probably reflects the fact that this toxin lacks critical amino acid residues responsible for nAChR binding [66]. Further studies will be needed to understand the biological functions of sNXH1 and its role in *M. corallinus* envenomation.

In contrast, sNXH8 caused a rapid, potent neuromuscular blockade through interaction with nAChRs that was completely reversed by washing the PND preparations (Figure 9a), or by adding the acetylcholinesterase inhibitor neostigmine (Figure 9b) or the Ca^2+^-dependent K^+^-channel blocker 3,4-DAP (Figure 9c). These findings indicated that sNXH8 acted postsynaptically on nAChRs. The sNXH8-induced neuromuscular blockade was neutralized by pre-incubation with coralsnake antivenom (Instituto Butantan; Figure 9d) or *O. scutellatus* (coastal taipan) antivenom (Figure 9e), corroborating the previously described cross neutralization of coralsnake venoms by Australian snake antivenoms [91].

The ease with which sNXH8-induced blockades could be reversed by washing the preparations or by the addition of neostigmine or 3,4-DAP indicated that the binding of this toxin to nAChRs was very weak (Figure 9a–c). This finding agrees with the in silico characterization of NXH8 as a non-conventional 3FTx, with a lower toxicity (LD_50_ of 5–80 mg/kg, i.v.) compared to more toxic α-neurotoxins (LD_50_ of 0.04–0.3 mg/kg, i.v.) [57,92].

Recombinant NXH8 was inactive on nAChR in this assay, confirming that the recombinant expression and purification of rNXH8 resulted in a biologically inactive toxin, although the antiserum produced against rNXH8 prevented the interaction of *M. corallinus* venom with nAChRs, as judged by the attenuated displacement of [^125^I]-labeled α-bungarotoxin from these receptors (Figure 8). The finding that only sNXH8 interacted with nAChRs (with rNXH8 being inactive) (Figure 9) highlights the usefulness of chemical synthesis for studying the pharmacological roles of 3FTx in animal venoms.

## 3. Discussion

Coralsnakes (*Micrurus* spp.) are the most abundant representatives of the Elapidae in the Americas [3,4]. The venoms of these snakes contain a variety of toxins, including PLA_2_ and 3FTx, the two most abundant groups. Numerous studies have shown that 3FTx exhibit a wide range of biological activities, such as the blockade of γ-aminobutyric acid (GABA) ionotropic receptors [93,94,95], interaction with interleukin or insulin receptors [96,97,98], and the activation of sperm motility [99]. Since 1963, with the characterization of the first 3FTx, α-bungarotoxin [66], nearly 1000 3FTx amino acid sequences have been deposited in the UniProt database, although few of these relate to toxins from *Micrurus* venoms.

Since 1963, with the characterization of the first representative, α-bungarotoxin [66], nearly a thousand 3FTx amino acid sequences have already been deposited in the UniProt database. Nevertheless, due to difficulties in venom extraction and its limited availability, few of these studies describe the characterization of toxins from the venom of *Micrurus* spp. snakes.

In the present study, two potential 3FTx (NXH1 and NXH8) from the venom of *Micrurus corallinus* were characterized. A cluster analysis of multiple sequence alignment showed that NXH1, which has been previously shown to be specific to *M. corallinus* venom [76,87], is a divergent polypeptide located between short α-neurotoxins and fasciculins and is not related to either NXH8 (3NO48_MICCO) or α-neurotoxins. It is likely to form a new class of three-finger toxins without cholinergic action (Figure 5).

Compared to short α-neurotoxins, although NXH1 lacks critical amino acid residues for nAChR binding, it exhibits a sequence similarity of 58.4% with Cobrotoxin b from *Naja atra* (3S1CC_NAJAT), 53.2% with short neurotoxin 1 from *Naja oxiana* (3S11_NAJOX), and an equal match of 53.2% with 3FTx Mnn I from *Micrurus nigrocinctus*. This similarity to short α-neurotoxins rises to 60.7% when compared with those from *Hydrophis cyanocinctus* (3S11_HYDCY), the Asian annulated sea snake. Additionally, NXH1 exhibits a comparable level of sequence similarity with toxins from the fasciculin family, showing a 51.6% similarity to Fasciculin-1 from *Dendroaspis angusticeps* and 46.3% to anti-acetylcholinesterase toxin C from *Dendroaspis polylepis polylepis* (Figure 3 and Figure 4).

NXH8 exhibits a fifth disulfide bond at the tip of its first loop, a feature also seen in several long α-neurotoxin homologues from *Bungarus* spp. as well as in some weak neurotoxins from *Naja* spp. One of these toxins with this additional disulfide bond is Bucandin (3NOJ_BUNCA), whose three-dimensional structure has been determined [100,101]. The Bucandin-type toxins, which are quite distinct from NXH8 with less than 30% sequence identity, constitute the most variable subgroup. Notably, Bucandin (3NOJ_BUNCA), isolated from *B. candidus* venom, is hypothesized to promote the release of acetylcholine [100,102]. The authors suggest that it acts as a presynaptic neurotoxin, though the biological assays to support this have not been published. This is notable because to date, all known neurotoxins in the family have been characterized as postsynaptic. Presynaptic activity has also been attributed to elapid venoms, but it was previously thought to be exclusively due to β-neurotoxins that share the structural pattern of phospholipase A2 (PLA2). Subsequent experiments have shown that Bucandin was non-lethal to mice even at elevated doses (50 mg/kg), casting doubt on its potential curaremimetic effects [103]. Furthermore, computational modeling investigating the binding interactions of various weak toxins with nicotinic acetylcholine receptors indicates instability in Bucandin’s complex formation with nAChR [104]. 

An analysis of NXH8 revealed that it is a polypeptide with a primary structure that contains 10 cysteine residues and potentially harbors a fifth disulfide bond in the first loop (Figure 1). Along with the non-conventional neurotoxins of Orphan group IV from *Bungarus* spp. venom [36], NXH8 is closely related to the short/long α-neurotoxins (Figure 5)—a non-homogeneous group that encompasses a spectrum of diverse functional activities [105]. 

Candoxin (3NO4_BUNCA), another representative of non-conventional neurotoxins from Orphan group IV, serves as a reversible antagonist to *Torpedo* and skeletal muscle nAChRs, and exhibits poor reversibility as an antagonist of neuronal α7 nicotinic acetylcholine receptors [60,61]. In addition, other non-conventional neurotoxins, from *Bungarus* spp. (categorized in Orphan group V [36]), exhibit a remarkable range of functions. For example, γ-Bungarotoxin (3NO5I_BUNMU) inhibits the binding of a specific muscarinic ligand to brain membranes, indicating a direct interaction of the toxin with the M2 receptor [106].

In addition to the *Bungarus* genus, non-conventional neurotoxins possessing a distinctive fifth disulfide bridge between Cys^6^–Cys^11^ have been identified in venoms and venom-gland cDNAs from *Naja* spp. These neurotoxins exhibit considerably lower similarity with those from *Bungarus* spp. and *M. corallinus* NXH8. Termed “weak toxins” [54,55,56,57,58,59], they demonstrate minimal toxicity upon mouse injection and are classified within Orphan group II [36]. Examples include NNAM3 from *Naja atra* (3NO23_NAJAT), which impedes cholinergic transmission in frog muscle at micromolar concentrations [102], weak neurotoxin 6 from *Naja naja* (3NO26_NAJNA) [59], and weak toxin CM-11 from *Naja haje haje* (3N02B_NAJHH) [107]. WTX—a weak tryptophan-rich neurotoxin isolated from *Naja kaouthia* (3NO2_NAJKA) [57]—and Wntx-5 from *Naja sputatrix* (3NO25_NAJNA) [108] bind to muscular receptors with low affinity and demonstrate an even lower affinity for neuronal (α7) nAChRs. Interestingly, WTX appears to influence hemodynamic regulation through its action on acetylcholine receptors [109]. NXH8 and its homologues from the *Bungarus* spp. exhibit a closer relationship to short/long α-neurotoxins (up to 68.18% similarity) compared to these weak neurotoxins from *Naja* spp. (Figure 2 and Figure 5).

Similarly, SLURP peptides, also belonging to the Ly-6/uPAR family, have been identified as modulators of nicotinic acetylcholine receptors (nAChRs). These peptides are known to influence various physiological processes, including immune response and neuronal signaling, by interacting with nAChRs. 

Given that Lynx-1, snake neurotoxins, ODR-2, and SLURP peptides exhibit substantial differences in their primary structures, it seems plausible that ODR-2 and SLURP peptides could also serve as cholinergic neuromodulators, similar to Lynx-1 and neurotoxins. These observations lead to the hypothesis that the pattern of five disulfide bridges may be related to a distinctive function involving the recognition of new types of receptors, suggesting that the integration of cholinergic modulation by three-finger proteins could be an ancient trait, and that the five-disulfide bond topology could be evolutionarily linked to an endogenous ligand for neuronal nAChR.

It is noteworthy that atoxic proteins from different animals with the three-finger structural motif also have the same fifth disulfide bridge at the tip of the first loop. Examples include the frog xenoxins [110,111], mammalian GPI-anchored membrane receptors such as CD59 and Ly-6 [112,113], the urokinase-type plasminogen activator receptor (uPAR) [114], ODR-2, a protein required for olfaction in *C. elegans*, exclusively expressed in cholinergic motor neurons [115], and Lynx-1, a neuromodulatory peptide that is highly expressed in Purkinje cells, binding with high affinity to homopentameric neuronal receptors. In vitro, the soluble form of Lynx-1 shows presynaptic activity by binding to α7-type receptors, enhancing acetylcholine release [116]. Similarly, SLURP peptides, also belonging to the Ly-6/uPAR family, have been identified as modulators of nAChRs. These peptides are known to influence various physiological processes, including immune response and neuronal signaling, by interacting with nAChRs.

Given that Lynx-1, snake neurotoxins, ODR-2, and SLURP peptides exhibit substantial differences in their primary structures, it seems plausible that ODR-2 and SLURP peptides could also serve as cholinergic neuromodulators, similar to Lynx-1 and neurotoxins. These observations lead to the hypothesis that the pattern of five disulfide bridges may be related to a distinctive function involving the recognition of new types of receptors, suggesting that the integration of cholinergic modulation by three-finger proteins could be an ancient trait, and that the five-disulfide bond topology could be evolutionarily linked to an endogenous ligand for neuronal nAChR [105,117].

Note, however, that the similarities of NXH8 to other polypeptides of known function do not necessarily imply it will have the same activity. As a matter of fact, the alignment of short and long chain α-neurotoxins with toxins of completely different pharmacological actions, such as cardiotoxins, reveals a set of invariant residues. This suggests that these residues are probably related to the structural integrity of the three-finger toxin fold.

On the other hand, both short and long chain α-neurotoxins have well-defined activities, providing a baseline for comparative studies of NXH8 and other related toxins. In fact, there are certain amino acid residues exclusively conserved in α-neurotoxins that are not found in cardiotoxins. The presence of such residues at very specific positions is particularly relevant, as it may suggest a role in the binding affinity to nAChRs [39,82,118]. In this manner, the alignment of NXH8 with short α-neurotoxins, like Erabutoxin-a from *Laticauda semifasciata* (3S1EA_LATSE) and NMM I from *Naja mossambica* (3S11_NAJMO), as well as with long α-neurotoxins such as α-Bungarotoxin from *B. multicinctus* (3L21A_BUNMU) and α-Cobratoxin from *Naja kaouthia* (3L21_NAJKA), shows that NXH8 possesses key residues necessary for skeletal muscle nAChR binding [39,119].

Particularly noteworthy is the conservation of important amino acids in NXH8 (Figure 2), located in Loop II of α-neurotoxins, a region harboring critical residues for receptor binding. Trp^31^, notably absent in NXH1 (Figure 3), along with Arg^39^, are key to the binding specificity of α-neurotoxins [82,118]. NXH8’s His^35^, differing from the typical Phe in similar toxins, aligns with histidine found in other potent α-neurotoxins, indicating a unique role in receptor interaction [39]. Mutagenesis studies on α-Cobrotoxin and NMM I have demonstrated that mutating Lys^29^ to Glu results in a differential decrease in binding affinity at two muscle nAChR binding sites. This charge inversion more significantly affects the α1/γ site compared to the α1/δ site [80,81,82]. The presence of Glu at position 29 in Candoxin suggests a differential selectivity for these sites [60], a characteristic potentially shared by NXH8, which exhibits Thr at this position.

In NXH8, the presence of Asn^33^ instead of Asp^33^ may not significantly impact receptor binding. A 46-fold reduction in receptor affinity is observed when an equivalent Asp^33^ (in Erabutoxin-a) is changed to His, whereas its replacement with asparagine marginally increases the Kd of Erabutoxin-a from 0.07 nM to 0.10 nM [79]. NXH8 also incorporates Arg^42^, a residue conserved in selected short neurotoxins and ubiquitously in long α-neurotoxins, essential for receptor binding in α-Cobratoxin [82]. A substitution at a comparable site in Erabutoxin-a, changing Ile to Arg, enhances its binding affinity for muscular nAChR [79]. Furthermore, Asp^45^ in NXH8 may also be involved in receptor binding [79,82,118], with analogous positions in Erabutoxin-a and α-Cobratoxin—glutamic and aspartic acids, respectively—being implicated in binding. Site-directed mutagenesis at this site has been shown to decrease affinity to the nicotinic receptor [82,118] (Figure 2).

Significant amino acid residues outside Loop II are present in NXH8’s primary structure, including Lys^56^ in Loop III (Figure 2). This residue, also found in NMM I, Erabutoxin a, α-Cobrotoxin, and α-Bungarotoxin, is critical for receptor binding [79,82]. In the first loop, critical residues like His^8^, Gln^9^, Ser^10^, and Gln^12^ are exclusive to short α-neurotoxins. 

Phe^8^, present in both NXH8 and Candoxin, may have a functional role similar to His^8^ in short α-neurotoxins or Phe/His^73^ in long α-neurotoxins, as the spatial disposition of Phe^8^ in Candoxin closely resembles Phe^73^ in α-Cobratoxin [105]. Additionally, Thr^10^, known to interact with Torpedo nAChRs in α-Bungarotoxin [84], is found in the same position in both NXH8 and α-Cobrotoxin/α-Bungarotoxin (Figure 2).

The primary structure of NXH1 (Figure 3), as seen in Candoxin, contains residues exclusive to short α-neurotoxins, such as His^8^ and Ser^10^. However, Phe/His^73^, located in the C-terminal tail, is unique to long neurotoxins [118].

Long-chain α-neurotoxins, like α-Bungarotoxin and α-Cobrotoxin, exhibit high affinity for both neuronal α7 and skeletal muscle nAChRs, characterized by a core structure with additional residues specific to each receptor subtype [51,118]. NXH8, similar to weak neurotoxins from *Naja* spp. and non-conventional neurotoxins from *Bungarus* spp., lacks certain invariant residues, suggesting potential low toxicity. Notably, NXH8 contains the functional motif common to both long and short chain α-neurotoxins for skeletal muscle nAChR binding (Figure 2), indicating possible affinity for these receptors or a unique epitope present in α-neurotoxins from *M. corallinus* venom, and potentially in other elapids like *M. altirostris*, *D. angusticeps*, and *B. multicinctus*. This epitope might explain why *M. corallinus* venom reduced α-Bungarotoxin’s binding to nAChRs, hinting at an antagonistic component, likely NXH8 (Figure 8).

The preincubation of anti-rNXH8 antibodies with the venom fully restored α-Bungarotoxin binding, implicating NXH8 as the antagonistic component. This effect was specific to anti-rNXH8 and not observed with anti-rNXH1 antibodies, likely due to NXH1 lacking several critical amino acid residues for nAChR binding (Figure 8).

Further, the study explored whether biologically active toxins could be synthesized chemically. Only synthetic NXH8 (sNXH8) showed antagonist activity on nAChRs in phrenic nerve–diaphragm preparations. Neither recombinant NXH8 (rNXH8) nor synthetic NXH1 (sNXH1) exhibited such activity (Figure 9), reversible with 3,4-DAP and neostigmine. Neutralization was also observed with commercial equine anti-elapidic serum [23] and anti-*Oxyuranus scutellatus* serum [91], suggesting structural similarities between NXH8 and *O. scutellatus* venom toxins.

The lack of antagonist activity in rNXH8 suggests possible incorrect protein folding, while sNXH1’s lack of activity is likely due to missing critical residues necessary for interactions with both short- and long-chain α-neurotoxins and skeletal muscle receptors, as well as long-chain neurotoxin binding to neuronal α7 receptors. NXH1’s classification in Orphan Group XII [36], which is positioned near a class of 3FTx with anticholinesterase activity, hints at an unknown biological function, warranting further investigation.

In conclusion, this study enhances our understanding of *M. corallinus* venom components and their pharmacological effects, particularly highlighting NXH8’s role as a weak nAChR antagonist at neuromuscular junctions, potentially influencing clinical outcomes of *M. corallinus* envenomation.

## 4. Materials and Methods

### 4.1. Venoms, Toxins, and Antivenom

Most of the venoms described here were obtained from the Instituto Butantan (São Paulo, SP, Brazil) except for those from *M. frontalis*, *M. surinamensis*, *M. carinicauda dumerilli*, *Dendroaspis angusticeps*, *Notechis scutatus scutatus*, and *Bungarus multicinctus* venoms that were purchased from Sigma-Aldrich/Merck (Darmstadt, HE, Germany). Anti-immunoconjugates and cardiotoxin from *Naja naja kaouthia* were also purchased from Sigma. Coralsnake antivenom produced by immunizing horses with *M. corallinus* and *M. frontalis* venoms at a 1:1 ratio was obtained from the Instituto Butantan [120]. Antivenom against *Oxyuranus scutellatus* (coastal taipan) venom was obtained from CSL (Melbourne, Australia).

### 4.2. cDNA Cloning, Recombinant Protein Expression, and Metal Ion Affinity Purification

The cloning of the *nxh1* nucleotide sequence (EMBL accession number: AF197563), as well as the expression and purification of recombinant NXH1 (rNXH1), were carried out as previously described [76]. The procedures described below refer specifically to NXH8.

#### 4.2.1. Cloning of nxh8

The nxh8 (EMBL accession number: AJ344067) nucleotide sequence was obtained from a phage cDNA library [74,75]. The nxh8 cDNA fragment was released from the phage DNA by *Eco*RI and *Not*I double digestion and was subcloned at the *Hinc*II site of pGEM3Zf(+) (Promega Corporation, Madison, WI, USA) by blunting the ends with DNA Polymerase I Large (Klenow). DNA sequencing confirmed that the *nxh8* fragment had been successfully subcloned in a reverse orientation.

For expression, the nxh8 cDNA fragment was amplified by PCR and was subcloned at the *Bam*HI site of the pRSET-C vector (Thermo Fisher Scientific, Waltham, MA, USA), which allows the expression of a 6x His-tagged recombinant protein at the N-terminus. This strategy used the M13 forward primer combined with a specific oligonucleotide (5’—c**gg atc c**tt gaa tgt aag ata tgc aac ttc—3′) that includes an upstream *Bam*HI restriction site (in bold) in frame with the first nucleotides of the mature NXH8 peptide coding sequence. The correct orientations of the clones were evaluated by double *Bam*HI/*Xba*I digestion and restriction analysis and were confirmed by DNA sequencing.

#### 4.2.2. DNA Sequencing

DNA sequencing was performed using the chain-terminator method [121]. The reactions were run in a PTC-100 thermal cycler (Bio-Rad Laboratories, Hercules, CA, USA) using a BigDye Terminator v3.1 Cycle Sequencing Kit (Thermo Fisher Scientific, Waltham, MA, USA), according to the manufacturer’s instructions. After purification, DNA sequence analysis was performed using an ABI Prism 3100 Genetic Analyzer (Thermo Fisher Scientific, Waltham, MA, USA).

#### 4.2.3. Recombinant Expression of NXH8

Chemically competent *E. coli* BL21 (DE3) cells were transformed with the pRSET-C-nxh8 plasmid using standard protocols [122]. Transformed cells were cultured at 37 °C, under constant agitation (200 rpm), in ampicillin (100 μg/mL)-supplemented Luria Broth (LB). Recombinant protein expression was induced by adding IPTG (1 mM) when the cell culture optical density at 600 nm (OD_600nm_) reached 0.6.

#### 4.2.4. Cell Lysis and Recombinant Protein Solubilization

After 3 h of induction, the cells were harvested by centrifugation (1000× *g*, 4 °C, 20 min) and resuspended in 50 mM Tris, pH 8.0, containing 100 mM NaCl, 10 mM EDTA, 1 mM PMSF, and 0.1% Triton-X 100. Cell lysis was achieved by sonication (3 × 1 min cycles, 60% amplitude, 20 kHz frequency, duty cycle of 0.5), with the samples kept in an ice bath during the process. Soluble and insoluble protein fractions were separated by centrifugation (12,000× *g*, 4 °C, 20 min).

#### 4.2.5. Solubilization of Inclusion Bodies

Recombinant NHX8 (rNXH8) was expressed as inclusion bodies that were resuspended in 50 mM Tris (pH 8.0), containing 8 M urea, 100 mM NaCl, and 10 mM 2-mercaptoethanol (2-ME). The samples were incubated on a tube roller for 1 h at room temperature and subsequently centrifuged (10,000× *g*; 4 °C; 15 min). The resulting supernatant was reserved for the refolding process.

#### 4.2.6. Renaturation of Recombinant Proteins

After solubilizing the inclusion bodies, the recovered recombinant proteins were renatured by dilution (1:200, *v*/*v*) in 100 mM Tris, pH 8.0, containing 150 mM NaCl, and 2 mM GSH-GSSG. The diluted protein solution was then kept under magnetic agitation for 10 h at 14 °C. Any protein aggregates were removed by centrifugation (12,000× *g*; 4 °C; 15 min) or by filtration through a 0.22 μm filter.

#### 4.2.7. Metal Ion Affinity Chromatography Purification

Renatured recombinant proteins were applied to a Ni^2+^-charged chelating resin for metal-ion chromatography (Cytiva, Marlborough, MA, USA). The resin was washed with five column volumes (cv) of 50 mM Tris, pH 6.3, containing 5 mM imidazole, and 100 mM NaCl. Protein elution was first achieved with 5 cv of 50 mM Tris, pH 8.0, containing 250 mM imidazole, and 100 mM NaCl, and then with 5 cv of 50 mM Tris, pH 8.0, containing 100 mM EDTA, and 100 mM NaCl.

Alternatively, rNXH8 was purified under denaturing conditions, with previously solubilized inclusion bodies being directly applied to a Ni^2+^-charged chelating resin for metal-ion chromatography. For this, the column was first equilibrated with 5 cv of 50 mM Tris, pH 8.0, containing 8 M urea, 100 mM NaCl, and 10 mM 2-ME, and washed with 5 cv of the wash buffer (see above) containing 4 M urea. Recombinant proteins were eluted with 5 cv of the elution buffer (see above) containing 2 M urea.

#### 4.2.8. Protein Dialysis

For recombinant proteins that were previously renatured, the eluted fractions were dialyzed against 10 volumes of phosphate-buffered saline (PBS) containing 5 mM EDTA, using semipermeable membranes with a nominal cutoff of 3500 Da. Four buffer changes were conducted at 4 °C at 12 h intervals. For recombinant proteins purified under denaturing conditions, the rNXH8 concentration was lowered to 0.1 mg/mL to reduce protein aggregation during dialysis against 10 volumes of PBS buffer containing 2 M urea, 2 mM GSH-GSSG, 10 mM 2-ME, and 5 mM EDTA. After 12 h of dialysis at 4 °C, 50% of the buffer was replaced with PBS containing 5 mM EDTA, resulting in four successive buffer changes.

### 4.3. Peptide Synthesis, Disulfide Bond Formation, and Protein Purification

The complete mature sequences of the NXH1 and NXH8 polypeptides were synthesized on a 50 mmol scale using Fmoc chemistry on a Prelude synthesizer (Gyros Protein Technologies, Tucson, AZ, USA) using 4-(4-Hydroxymethyl-3-methoxyphenoxy)butyric acid (HMPB) ChemMatrix resin (Biotage Corporation, Uppsala, Sweden) functionalized with the appropriate protected amino acid. 

Fmoc-protected amino acids were used with the following side-chain protections: t-butyl ester (Glu,Asp), t-butyl ether (Ser, Thr, Tyr), trityl (Cys, His, Asn, Gln), 2,2,5,7,8-pentamethyl-chromane-6-sulfonyl (Arg), and t-butyloxycarbonyl (Trp). Each coupling step was carried out twice for 5 min using a mixture of 10:10:20 equivalents of Fmoc-amino acid/O-(1H-6-Chlorobenzotriazole-1-yl)-1,1,3,3-tetramethyluronium hexafluorophosphate (HCTU)/N-Methylmorpholine (NMM) in N-Methylpyrrolidone (NMP) (Sigma-Aldrich/Merck, Darmstadt, GER), followed by a capping step. Fmoc deprotection was then conducted using a 20% piperidine solution in NMP. After concomitant cleavage from the resin and amino acid side-chain deprotection using a solution of trifluoroacetic acid (TFA), H_2_O, and triisopropylsilane (TIS) in a 90:5:5 ratio, the peptides were precipitated in cold ethanol (EtOH), lyophilized from a 10% acetic acid solution, and purified by reverse-phase semi-preparative HPLC using a 250 × 10 mm Vydac C18 column (Vydac, Hesperia, CA, USA) with a gradient of 40–60% of 60% acetonitrile and 0.1% TFA over 40 min. 

The peptides were subjected to folding in Tris–HCl 100 mM, pH 8.0, containing 1 mM of EDTA, 1 mM GSH-GSSG, and 20% (*v*/*v*) glycerol at a peptide concentration of 0.05 mg/mL. After incubation for 24–36 h at 4 °C, the pH was lowered to 3 by the addition of 30% TFA, and the final reticulated peptides were purified using the same method described above and subsequently characterized by ESI-MS with a Bruker Esquire HCT 3000 plus instrument (Bruker Corporation, Billerica, MA, USA) for a final confirmation of the structure and the purity of the synthesized peptides. To minimize protein adsorption on the vessel, the folding process was conducted in 10 mL Microsorb tubes (Nunc ^TM^, Thermo Fisher Scientific, Waltham, MA, USA).

For clarity, the synthetic toxins are referred to as sNXH1 and sNXH8 to differentiate them from their recombinant counterparts.

### 4.4. Sequences, Alignments, and Analysis

A computer search was run using the PSI-BLAST [123] algorithm in the GenBank and EMBL/UniProt databases. Local alignments were conducted using the MACAW program with the BLOSUM62 matrix [124] and a sequence comparison dendrogram was constructed using PHYLIP—Phylogeny Inference Package [125]. Maximum likelihood estimates were used to calculate the distance between protein sequences, with a model based on the constraint of changing to different physicochemical properties of amino acid categories: (I, L, V, M), (F, W, Y), (K, R, H), (D, N, E, Q), (G, A, S, T), (P), and (C). Gaps were treated as substitutions. The probability of a category change was 0.55. The distance matrices were used to construct functional trees with the neighbor-joining method. A copy of the complete sequence alignment can be obtained by contacting the corresponding authors.

### 4.5. Antibody Production and Western Blotting

Purified recombinant proteins were used to immunize mice through intraperitoneal injections of 10 µg of protein in an alum suspension (1:1 ratio, *v*/*v*) administered at 14-day intervals. The serum antibody titer was monitored using ELISA, as described elsewhere [122]. Antiserum against rNXH8, recombinant NXH1 (rNXH1), and a mixture of both recombinant toxins (1:1 ratio, anti–MIX) was also obtained. For Western blotting, proteins were separated by SDS-PAGE (10–20% gradient acrylamide/bisacrylamide gel) under reducing conditions. The separated proteins were then transferred onto a nitrocellulose membrane and the immunoreactions were performed as previously described [126]. The resulting immunoreactive bands were visualized using an immunoperoxidase conjugate and a diaminobenzidine-, H_2_O_2_-, and CoCl_2_-staining procedure [126].

### 4.6. Acetylcholine Receptor Binding Assay

Cell membranes from primary skeletal muscle cells of neonatal rats were used for the nAChR-binding assay, as described by de Almeida-Paula, L.D. et al. [127]. The membranes were obtained by mechanically detaching cells from culture plates, followed by centrifugation and homogenization in a binding buffer (20 mM Tris-HCl, pH 7.4, containing 10 mM EDTA and 150 mM NaCl). The protein content was determined using the Bradford dye-binding method with bovine serum albumin (BSA) as the standard. The membrane preparations were stored at −70 °C until use.

For the binding assays, 25 μg of the membrane preparation was used for each reaction in the presence of 0.2 mg/mL of BSA. Individual membrane preparations were incubated at room temperature with different substances, including saline, different concentrations of *M. corallinus* venom (0.05, 0.5, or 5 μg/100 μL reaction volume), anti-rNXH1 or anti-rNXH8 antibodies, or a combination of *M. corallinus* venom pre-incubated with 10 µL of either anti-rNXH1, anti-rNXH8, or anti-MIX sera. After 1 h of incubation, the membranes were washed with a BSA-containing binding buffer. To each reaction tube, (^125^I) α-Bungarotoxin (2 × 10^−9^ M, 146 Ci/mmol) was added as the nAChR tracer before incubation for an additional 1 h at room temperature. The membranes were then washed, harvested by centrifugation, and the radioactivity remaining in the pellet was counted. Nonspecific binding was defined as any binding that occurred in the presence of 1 × 10^−2^ M of nicotine.

### 4.7. Twitch Tension Experiments

Male BALB/c mice (25–30 g) obtained from the Multidisciplinary Center for Biological Investigation (CEMIB UNICAMP) were housed on a 12 h light/dark cycle (lights on at 6 a.m.) in plastic cages (5–10/cage) with a wood shaving substrate (changed twice a week) in ventilated stands (Alesco^®^, Monte Mor, SP, Brazil) at 23 ± 1 °C with free access to water and rodent chow (Nuvital^®^, Colombo, PR, Brazil). Phrenic nerve–diaphragm (PND) preparations obtained from isofluorane euthanized mice were mounted under a resting tension of 1 g in 5 mL organ baths containing aerated (5% CO_2_ and 95% O_2_) Tyrode solution (composition, in mM: 137 NaCl, 2.7 KCl, 1.8 CaCl_2_, 0.49 MgCl_2_, 0.42 NaH_2_PO_4_, 11.9 NaHCO_3_, and 11.1 glucose, pH 7.0) at 37 °C and were allowed to stabilize for 10 min prior to testing the toxins, as described elsewhere [88,89]. 

Supramaximal stimuli (0.1 Hz, 0.2 ms) were delivered to the nerve by a Grass S88 stimulator (Grass Instrument Co., Quincy, MA, USA), and the muscle twitches were recorded using transducers, amplifiers, and LabChart v.7 software from ADInstruments (Bella Vista, New South Wales, Australia). After stabilization, the preparations were incubated for 60 min (or until complete neuromuscular blockade) with 10 μg/mL of rNXH1, sNXH1, rNXH8, or sNXH8. To ensure that the direct stimulation of the nerve–muscle preparations did not contribute to the overall tension recorded, 40 mM MgCl_2_ was added to the bath, and in all cases, the indirectly evoked twitches were completely blocked. The MgCl_2_ was subsequently removed by successive washes until the twitch amplitude had returned to the basal level. 

The reversibility of the neuromuscular blockade was assessed by adding neostigmine (NEO, an acetylcholinesterase inhibitor; 29 μM) or 3,4-diaminopyridine (3,4-DAP, a potassium channel blocker; 230 μM) to the preparations after ≥90% blockade and assessing the recovery of the twitch tension. Additionally, the neutralization of the neurotoxicity of sNXH8 was assessed by pre-incubating the toxin (37 °C, 30 min) with coralsnake antivenom (Instituto Butantan), or antivenom to the venom of *Oxyuranus scutellatus* (coastal taipan). Similar experiments were not conducted with sNXH1 since this toxin did not cause a neuromuscular blockade (see Section 2).

### 4.8. Ethics Statement

The animal experiments described in this report were approved by the Committee for Ethics in Animal Use of the Universidade Nove de Julho (CEUA/UNINOVE, protocol no. 4463100419). The experiments were conducted according to the general ethical guidelines for animal use established by the Brazilian Society of Laboratory Animal Science (SBCAL) and Brazilian legislation (Federal Law no. 11,794, of 8 October 2008), in conjunction with the guidelines for animal experiments established by the Brazilian National Council for the Control of Animal Experimentation (CONCEA) and EU Directive 2010/63/EU for the Protection of Animals Used for Scientific Purposes.

## Figures and Tables

**Figure 1 toxins-16-00164-f001:**
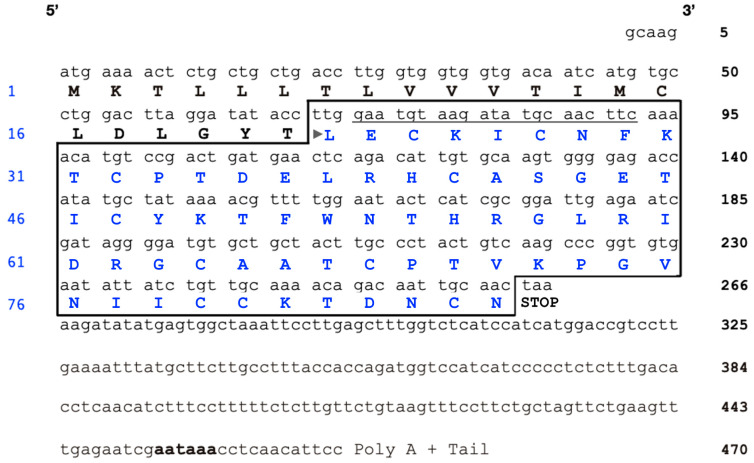
The nucleotide and deduced protein sequence of the nxh8 cDNA clone. The cDNA clone structure of the nxh8 gene from the venom gland of *Micrurus corallinus* (EMBL data bank accession number AJ344067) comprises the following regions: a 5′ untranslated region (UTR) (1–5 bp), a signal peptide coding sequence (6–68 bp), a mature peptide coding sequence (69–263 bp, highlighted in a box), and a 3′UTR (264–470 bp). The polyadenylation signal is indicated in bold. The deduced mature peptide sequence (NXH8) starts at the Leu residue and is indicated by an arrow. The sequence utilized for the 5′ primer design for PCR amplification and subcloning into the pRSET C expression vector is emphasized with an underline. The blue numbers on the left indicate amino acid residue positions, while the black numbers on the right indicate nucleotide positions.

**Figure 2 toxins-16-00164-f002:**
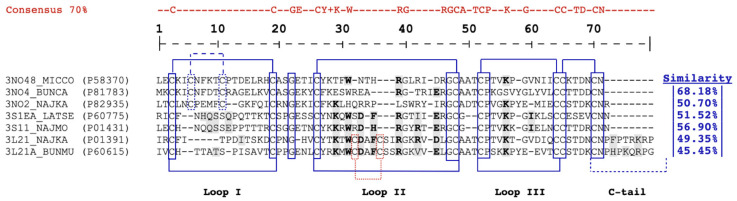
Alignment of NXH8 (3NO48_MICCO) with different Elapidae 3FTx highlighting functionally invariant residues related to skeletal muscle nAChR binding. Functionally invariant residues in three-finger α-neurotoxins that interact with muscular nicotinic acetylcholine receptors (nAChRs) are shaded in gray, based on experimental mutational data for Erabutoxin-a (3S1EA_LATSE) [78,79], NMM I (3S11_NAJMO) [80,81], and α-Cobratoxin (3L21_NAJKA) [82,83], as well as structural data for α-Bungarotoxin (3L21A_BUNMU) [84,85]. Residues critical for binding to skeletal muscle (α1)2β1γδ nAChRs, which are shared between short- and long-chain α-neurotoxins, are indicated in bold. Putative functional conserved residues in toxins such as WTX (3NO2_NAJKA), Candoxin (3NO4_BUNCA), and NXH8 (3NO48_MICCO), with analogous binding functions, are also presented. The cysteine residues and structurally invariant residues are enclosed in boxes, and the disulfide bridges are highlighted. Furthermore, the segments contributing to the three loops and the C-terminus are labeled. The fifth disulfide bridge in long-chain α-neurotoxins, between Cys^32^ and Cys^36^, is represented by a dotted line, while the corresponding bridge in non-conventional three-finger toxins, between Cys^6^ and Cys^11^, is shown as a dashed line.

**Figure 3 toxins-16-00164-f003:**
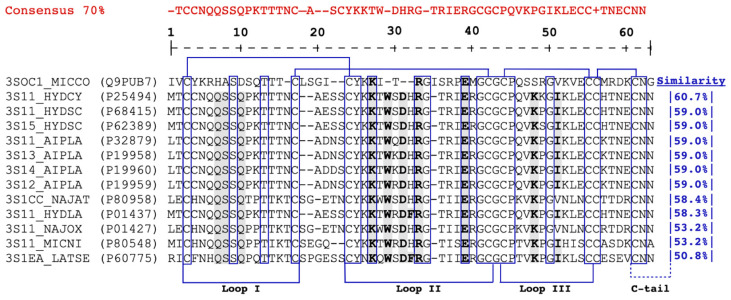
Alignment of NXH1 (3SOC1_MICCO) with different Elapidae 3FTx highlighting functionally invariant residues related to skeletal muscle nAChR binding. Functionally invariant residues in three-finger α-neurotoxins that interact with muscular nicotinic acetylcholine receptors (nAChRs) are shaded in gray, based on experimental mutational data for Erabutoxin-a (3S1EA_LATSE). Residues critical for binding to skeletal muscle (α)2βγδ nAChRs, which are shared between short- and long-chain α-neurotoxins, are indicated in bold. Putative functional conserved residues in toxins such as NXH1 from *M. corallinus* (3SOC1_MICCO), short neurotoxin 1 from *Hydrophis cyanocinctus* (3S11_HYDCY), short neurotoxins 1 from *Hydrophis schitosus* (3S11_HYDSC), Toxin 5, from *H. schitosus* (3S15_HYDSC), short neurotoxin A, from *Aipysurus laevis* (3S11_AIPLA), short neurotoxin B, from *A. laevis* (3S12_AIPLA), short neurotoxin c, from *A. laevis* (3S13_AIPLA), short neurotoxin D, from *A. laevis* (3S14_AIPLA), Cobrotoxin-b, from *Naja atra* (3S1CC_NAJAT), short neurotoxin 1, from *Hydrophis lapemoides* (3S11_HYDLA), short neurotoxins 1, from *Naja oxiana* (3S11_NAJOX), and three-finger toxin Mnn I, from *Micrurus nigrocinctus* (3S11_MICNI). The cysteine residues and structurally invariant residues are enclosed in boxes, and the disulfide bridges are highlighted. Furthermore, the segments contributing to the three loops and the C-terminus are labeled.

**Figure 4 toxins-16-00164-f004:**
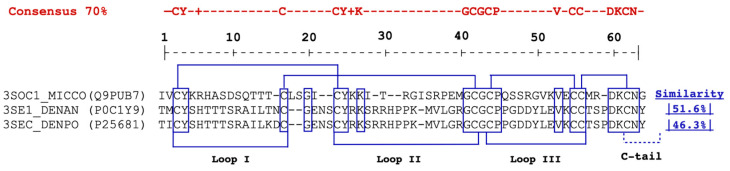
Alignment of NXH1 (3SOC1_MICCO) with two fasciculins 3FTx from *Dendroaspis* spp. Putative functional conserved residues in toxins such as NXH1 from *M. corallinus* (3SOC1_MICCO), Fasciculin-1, from *D. angusticeps* (3SE1_DENAN), Acetylcholinesterase toxin C, from *D. polylepis polylepis* (3SEC_DENPO). The cysteine residues and structurally invariant residues are enclosed in boxes, and the disulfide bridges are highlighted. Furthermore, the segments contributing to the three loops and the C-terminus are labeled.

**Figure 5 toxins-16-00164-f005:**
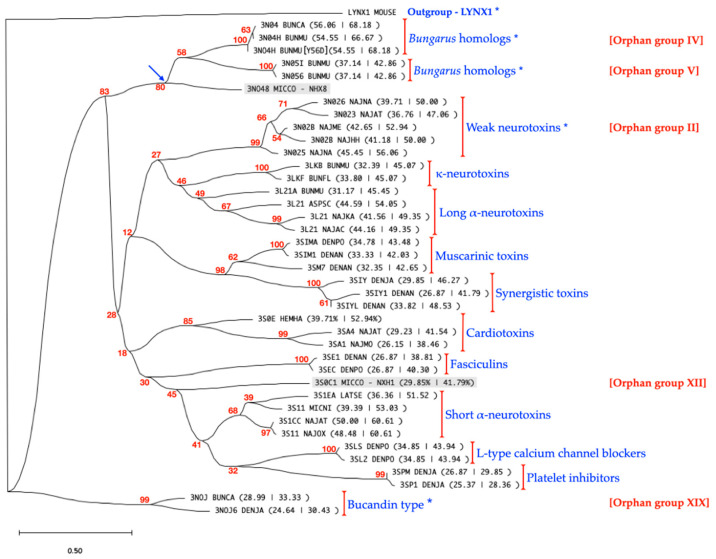
Dendrogram of three-finger toxins family. A blue arrow (➔) indicates the NXH8 ally group. 3FTx with a fifth disulfide bridge at loop I are indicated by a star (*). The dendrogram shows that 3FTx with a disulfide bridge at loop I is a non-homogeneous group, probably with distinct functions. The *M. corallinus* three-finger toxins NXH1 and NXH8 (shaded) are unrelated. Orphan groups’ clades are numbered in agreement with Fry, B.G. et al. [36]. The parentheses display, respectively, the identity and similarity percentages of each toxin in relation to NXH8. Bootstrap values are shown in red.

**Figure 6 toxins-16-00164-f006:**
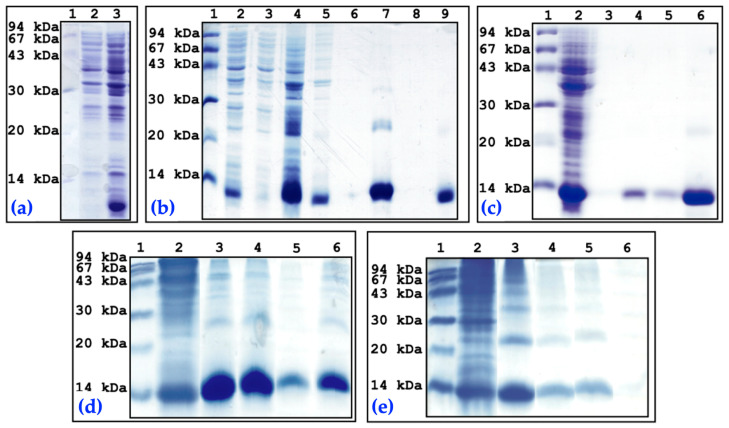
SDS-PAGE analysis of recombinant expression and purification of rNXH8 in *Escherichia coli* cells transformed with pRSETC-*nxh8* plasmid, with each lane loaded with a 10 μL sample volume. (**a**) *E. coli* BL21 (DE3)-pRSETC-nxh8 cell extracts: 1. Molecular-mass marker; 2. Cell extract before IPTG induction; 3. Cell extract after IPTG induction; (**b**) Metal ion affinity chromatography, under denaturing conditions (8 M urea/10 mM 2-ME), of rNXH8 from solubilized inclusion bodies: 1. Molecular-mass marker; 2. *E. coli* BL21 (DE3)-pRSETC-*nxh8* cell extract after IPTG induction; 3. Supernatant after cell lysate centrifugation; 4. Precipitate after cell lysate centrifugation; 5. Inclusion bodies after solubilization in 8 M urea/10 mM 2-ME buffer; 6. Non-adsorbed material (flow through), after sample loading; 7. Ni^2+^-charged resin after sample loading; 8. Non-adsorbed material after column wash; 9. Adsorbed material after elution; (**c**) Metal ion affinity chromatography of refolded rNXH8: 1. Molecular-mass marker; 2. Cell lysate extract; 3. Column wash with 5 mM imidazole; 4. Adsorbed material after elution with 1 column volume of 250 mM imidazole elution buffer; 5. Adsorbed material after elution with five column volumes of 250 mM imidazole elution buffer. 6. Adsorbed material after EDTA chelating elution; (**d**) Reducing SDS-PAGE analysis of rNXH8 aggregate formation during dialysis (2-ME was added to samples): 1. Molecular-mass marker; 2. Inclusion bodies after solubilization in 8 M urea/10 mM 2-ME buffer; 3. Adsorbed material after elution with 2 M urea/10 mM Imidazole buffer; 4. Full dialysis material; 5. Supernatant after dialysis material centrifugation; 6. Precipitate after dialysis material centrifugation. (**e**) Non-reducing SDS-PAGE analysis of rNXH8-aggregate formation during dialysis (2-ME was not added to sample): 1. Molecular-mass marker; 2. Inclusion bodies after solubilization in 8 M urea/10 mM 2-ME buffer; 3. Adsorbed material after elution with 2 M urea/10 mM Imidazole buffer; 4. Full dialysis material; 5. Supernatant after dialysis material centrifugation; 6. Precipitate after dialysis material centrifugation.

**Figure 7 toxins-16-00164-f007:**
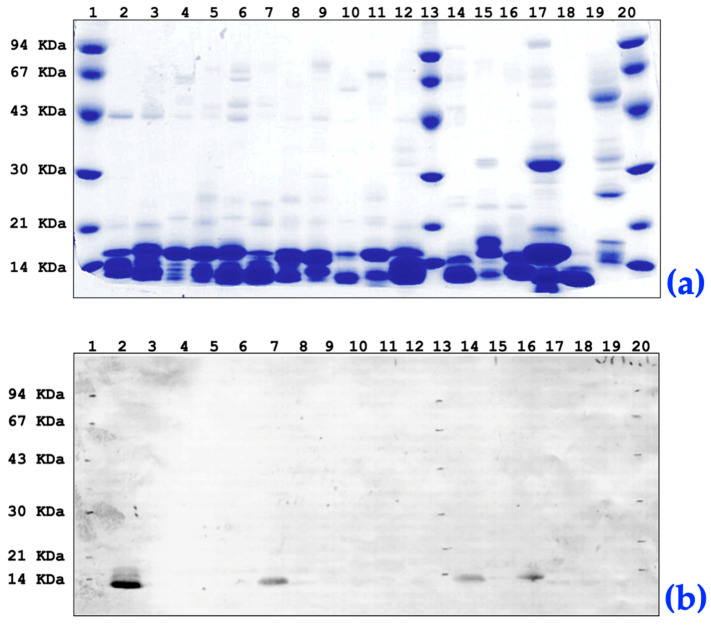
Cross-reactivity of anti-rNXH8 in diverse snake venoms by Western-blot. (**a**) SDS-PAGE stained with Coomassie Blue R250 (gradient 10–20%), with each lane loaded with a 10 μL sample volume. (**b**) Western blot of a replica gel after electroblotting to a nitrocellulose support incubated with anti-NXH8 polyclonal serum. 1. Molecular-mass marker; 2. Venom of *M. corallinus*; 3. Venom of *M. ibiboboca*; 4. Venom of *M. lemniscatus*; 5. Venom of *M. spixii*; 6. Venom of *M. frontalis*; 7. Venom of *M. altirostris*; 8. Venom of *M. surinamensis*; 9. Venom of *M. carinicauda dumerilli*; 10. Venom of *M. hemprichii*; 11. Venom of *M. spixii martiusi*; 12. Venom of *M. decoratus*; 13. Molecular-mass marker; 14. Venom of *Dendroaspis angusticeps*; 15. Venom of *Notechis scutatus scutatus*; 16. Venom of *Bungarus multicinctus*; 17. Venom of *Crotalus durissus terrificus*; 18. Purified cardiotoxin IV *Naja naja kaouthia*; 19. Venom of *Bothrops jararaca*. 20. Molecular-mass marker. The polyclonal serum against recombinant NXH8 reacts with homologous *M. corallinus* venom and with heterologous venoms from *M. altirostris* (lane 07), *Dendroaspis angusticeps* (lane 14) and *Bungarus multicinctus* (lane 16).

**Figure 8 toxins-16-00164-f008:**
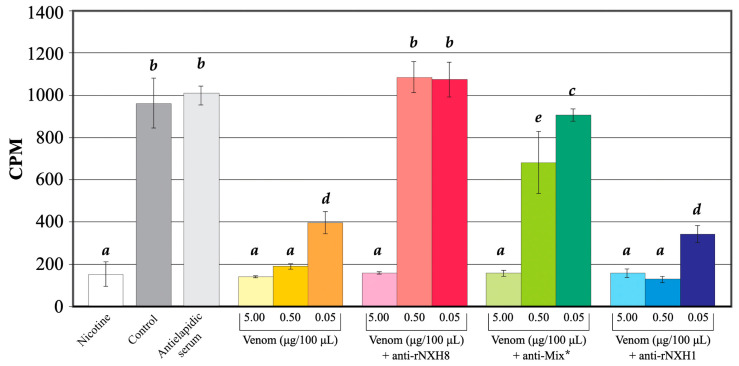
Acetylcholine receptor-binding assay utilizing primary skeletal muscle cell-membrane preparations from newborn rats. The assay employed [^125^I]-labeled α-Bungarotoxin as a tracer. Membrane preparations were incubated for one hour with either nicotine or the *M. corallinus* crude venom, followed by a wash in binding buffer, and then further incubated for an additional hour at room temperature with [^125^I]-labeled α-Bungarotoxin. Subsequently, the membranes were rewashed, and the resulting pellets, obtained by centrifugation, were analyzed for residual radioactivity. In the neutralizing venom assay, diminishing quantities of crude venom were pre-incubated for 30 min with 10 µL of polyclonal sera (anti-rNXH8, anti-rNXH1, or anti-MIX*) before the addition to the membrane preparation. Assays were performed in quadruplicate. Statistical analysis was conducted using a 2-way ANOVA with multiple comparisons in GraphPad Prism 10 (GraphPad Software, Boston, MA, USA), with a significance threshold of *p* < 0.05. Different lowercase letters (a–e) above the bars denote statistically distinct groups. Note: * Anti-MIX represents a 1:1 mixture of anti-rNXH8 and anti-rNXH1.

**Figure 9 toxins-16-00164-f009:**
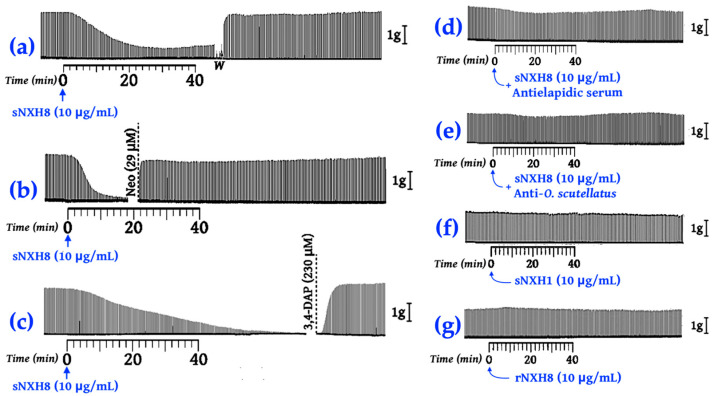
Representative recordings showing the neuromuscular activities NXH1 and NXH8 three-finger toxins in indirectly stimulated PND preparations at 37 °C. (**a**) Reversion of synthetic NXH8 (sNXH8) blockade by saline washing (*W*); (**b**) Reversion of sNXH8 blockade by neostigmine (29 μM); (**c**) Reversion of sNXH8 (10 μg/mL) blockade by 3,4-DAP (230 μM); (**d**) sNXH8 activity after pre-incubation (37 °C, 30 min, 1:1 *v*/*w* antivenom–toxin ratio) with antielapidic serum from *Instituto Butantan*; (**e**) sNXH8 activity after pre-incubation (37 °C, 30 min, 1:1 *v*/*w* antivenom–toxin ratio) with anti-*Oxyuranus scutellatus* (Coastal Taipan) serum; (**f**) Synthetic NXH1 (sNXH1) activity; (**g**) Recombinant NXH8 (rNXH8) activity.

## Data Availability

The data presented in this study are available on request from the corresponding author.

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
