# Peer review of "The Cloning and Characterization of a Three-Finger Toxin Homolog (NXH8) from the Coralsnake Micrurus corallinus That Interacts with Skeletal Muscle Nicotinic Acetylcholine Receptors"

_toxins, 2024, doi:10.3390/toxins16040164_

Round 1
Reviewer 1 Report
Comments and Suggestions for Authors
The author(s) characterized two previously identified 3FTx (NXH1 and NXH8) from Micrurus corallinus venom, employing in silico, in vitro, and ex vivo experiments. The study's findings contribute to our understanding of the structural diversity of 3FTx and the mechanisms underlying coral snake venom toxicity. The results are extensive, and the manuscript is well-prepared. I have only few comments on figure presentations and text editing, and once these are addressed, I believe the manuscript will be suitable for publication in Toxins.
1. In Figure 1, all genus and species names should be italicized throughout the manuscript.
2. In Figure 2, all amino acid symbols shaded in gray are also indicated in bold. Is this correct? Additionally, the dotted line and dashed line should be exchanged.
3. In Figure 3, all amino acid symbols shaded in gray are also indicated in bold. Is this correct?
4. In Figures 3 and 4, the C-terminus is not labeled.
5. In Figure 5, the identity percentages of 3C1CC NAJAT (50.00%) and 3S11 NAJOX (48.48%) in relation to NXH8 are not low, but these two toxins are distant from NXH8 in the dendrogram. Why? Additionally, the bootstrap values of the dendrogram and related statistics of the PHYLIP analyses need to be shown to support result reliability.
6. How much sample was added to each lane of each gel in Figures 6 and 7?
7. In Figure 7(a), lane numbers are misplaced, with only 19 shown although there are actually 20 lanes. Also, is it correct that an immunoreactive band was seen in M. altirostris venom (lane 7 in Figure 7(b))?
8. In Figure 8, the X-axis title should be revised. The sample size in each trial, the statistical results of multiple comparison, and the significance (e.g. P value) of the statistical tests need to be presented.
9. In Figure 9, why is there a gap (in the bottom line) during the administration of neostigmine and DAP in diagrams (b) and (c), respectively?
10. At line 16, the sentence should be revised.
11. At lines 72-73, ''inhibitors of platelet aggregation inhibitors'' should be revised.
12. At line 522, the sentence should be revised.
13. At lines 586-593, one paragraph needs to be removed.
14. At line 792, the accessed date is not shown.
15. The formats of all references need to be consistent. For example, the format of references 16, 19, 20, 30, 32, etc. differs from the others.
Author Response
Dear Reviewer.
Thank you for your diligent review of our manuscript. We appreciate the time and effort you have invested in providing your valuable feedback.
Below, we have addressed each of your comments in detail:
1. In Figure 1, all genus and species names should be italicized throughout the manuscript. R: As requested, all genus and species names have been italicized in Figure 1 and throughout the manuscript for consistency and adherence to scientific naming conventions.
2. In Figure 2, all amino acid symbols shaded in gray are also indicated in bold. Is this correct? Additionally, the dotted line and dashed line should be exchanged.
R: Your observation is correct; there were indeed some residues that should not have been bolded. Additionally, certain residues should also be highlighted in grey. The figure has been adjusted accordingly to reflect these corrections.
3. In Figure 3, all amino acid symbols shaded in gray are also indicated in bold. Is this correct?
R: Your observation is correct; there were indeed some residues that should not have been bolded. The figure has been adjusted accordingly to reflect these corrections.
4. In Figures 3 and 4, the C-terminus is not labeled.
R: Your observation is correct; the C-terminus tail was added, and both figures have been adjusted accordingly.
5. In Figure 5, the identity percentages of 3C1CC NAJAT (50.00%) and 3S11 NAJOX (48.48%) in relation to NXH8 are not low, but these two toxins are distant from NXH8 in the dendrogram. Why? Additionally, the bootstrap values of the dendrogram and related statistics of the PHYLIP analyses need to be shown to support result reliability.
R: Bootstrap values were added to the dendrogram. The observed distance in the dendrogram between NXH8 and the toxins 3C1CC NAJAT and 3S11 NAJOX, despite a 50% similarity, can likely be attributed to the unique structural features of NXH8. Specifically, NXH8 is characterized by the presence of 10 cysteine residues, forming an additional fifth disulfide bond located within the first loop. This structural characteristic is not shared by the toxins 3C1CC NAJAT and 3S11 NAJOX, which are classified as short α-neurotoxins containing 8 cysteine residues and four disulfide bonds. The 50% similarity may be a result of both classes of toxins sharing critical residues for binding to the nAChR. In the case of the aforementioned toxins, such residues might include Q7, S8, Q10, K26, W28, D30, H31, R32, E37, K46. These shared residues contribute to the functional similarities, whereas the distinct cysteine framework and the consequent disulfide bonding patterns contribute to the observed phylogenetic divergence in the dendrogram.
6. How much sample was added to each lane of each gel in Figures 6 and 7?
R: 10 μL. For clarity, this information was added in the descriptions of both figures.
7. In Figure 7(a), lane numbers are misplaced, with only 19 shown although there are actually 20 lanes. Also, is it correct that an immunoreactive band was seen in M. altirostris venom (lane 7 in Figure 7(b))?
R: We have corrected the lanes identification of both figures 7a and 7b. Yes, despite there was a misplacement on lanes, it is correct that an immunoreactive band was seen in M. altirostris venom (lane 7 in Figure 7b)
8. In Figure 8, the X-axis title should be revised. The sample size in each trial, the statistical results of multiple comparison, and the significance (e.g. P value) of the statistical tests need to be presented. R: Figure 8 was adjusted as requested. Sample size and statistical data were included. X-axis was revised for a better understanding, as well.
9. In Figure 9, why is there a gap (in the bottom line) during the administration of neostigmine and DAP in diagrams (b) and (c), respectively?
R: Actually, there is no gap. It is just the white background of the text boxes used to describe Neostigmine and 3,4-DAP.
10. At line 16, the sentence should be revised.
R: The sentence has been revised.
11. At lines 72-73, ''inhibitors of platelet aggregation inhibitors'' should be revised.
R: The sentence has been revised.
12. At line 522, the sentence should be revised.
R: The sentence has been revised.
13. At lines 586-593, one paragraph needs to be removed.
R: The last paragraph has been removed.
14. At line 792, the accessed date is not shown.
R: You are correct. To address this, we have chosen to archive the website, as suggested in the journal's instruction for authors, we have archived the website and adjusted this reference accordingly.
15. The formats of all references need to be consistent. For example, the format of references 16, 19, 20, 30, 32, etc. differs from the others.
R: References formats were updated to meet MDPI Toxins, as suggested in Journal's instructions.
Reviewer 2 Report
Comments and Suggestions for Authors
Snakebites are an easily neglected public health issue in tropical and subtropical regions of the world and antivenom is crucial first aid medicine for venomous snakebites. So, developing effective snake antivenom is very important to treat snakebites.
The present study expressed two 3FTs from the Coral snake Micrurus corallinus in E. coli and synthesized them as well. However, the neutralization effect of their poly antibodies was not very good.
1. In vivo survival tests using mice are recommended to be performed to evaluate the neutralization effect of the antibodies/antivenom against the natural venom of Coralsnake Micrurus corallinus.
2. Usually, the side effects of antivenom should not be ignored and It is recommended to produce F(ab’)2 antitoxins.
3. The structure of 3FTs has been previously reported a lot in PDB. It is better to analyze the structure of this 3FT and compare them with others as well.
Author Response
Dear Reviewer.
Thank you for your diligent review of our manuscript. We appreciate the time and effort you have invested in providing your valuable feedback.
Below, we have addressed each of your comments in detail:
1. In vivo survival tests using mice are recommended to be performed to evaluate the neutralization effect of the antibodies/antivenom against the natural venom of Coralsnake Micrurus corallinus.
R: Thank you for your suggestion. We understand the importance of in vivo studies in evaluating antivenom efficacy. Due to the limited quantity of Micrurus corallinus venom available for this study, we prioritized the structural characterization of the toxin and its potential interaction with nAChRs. However, we are in the process of planning a comprehensive in vivo study to assess the lethality of NXH8 and the neutralizing effectiveness of anti-sNXH8 antibodies against M. corallinus venom.
2. Usually, the side effects of antivenom should not be ignored and It is recommended to produce F(ab’)2 antitoxins.
R: We appreciate your emphasis on the potential side effects of antivenoms and the recommendation to produce F(ab')2 antitoxins. The development of F(ab')2 antitoxins, known for their reduced immunogenicity, is indeed a valuable consideration for enhancing antivenom safety. Your observation will be considered in the planning of our future projects.
3. The structure of 3FTs has been previously reported a lot in PDB. It is better to analyze the structure of this 3FT and compare them with others as well.
R: Your observation is indeed pertinent. In a previous study examining the neutralizing capabilities of specific NXH8 epitopes, we have already conducted a preliminary structural modeling analysis [Ref. 77]. Nonetheless, we are currently expanding on this with AI-powered in silico studies, which includes advanced modeling of NXH8 together with its docking with different biological targets. These findings are intended to complement the in vivo studies we are conducting and will be presented together in a future publication.
Reviewer 3 Report
Comments and Suggestions for Authors
The article entitled " Cloning and Characterization of a Three-Finger Toxin Homolog (NXH8) from the Coral snake Micrurus corallinus that Interacts with Skeletal Muscle Nicotinic Acetylcholine Receptors." has elucidated clone and recombinant expression of two toxins, NXH 1 and NXH 8. By using sequence analysis and a series of experiments, the biological activities of two toxins were predicted and evaluated. In some cases, these results provide more information for our understanding of M.corallinus venom.
General comments:
1. For cysteine-rich polypeptide toxins, it is theoretically possible to form a variety of isomers. Whether chemically synthesized or recombinant peptides, how can the authors identify the disulfide bridge of the final product?
2. In the Materials and Methods section, more details should be given on peptide synthesis and oxidative folding procedures. For example, Cysteine protecting groups (Acm or Trt), and the step of oxidative folding. The in vitro oxidative folding of disulfide-rich proteins can be challenging.
3. The recombinant 6xHis-NXH8 fusion protein has a theoretical molecular mass of 11,399 Da. However, the SDS-PAGE results alone were not sufficient to verify the correctness of the recombinant protein. HPLC and mass spectrometry are recommended for purification and identification of recombinant proteins.
4. Although studies have shown that NXH8 interacts with skeletal muscle nAChRs, its specificity, and selectivity for other subtypes of nAChRs were not evaluated.
5. Statistical analysis and comparison of different concentration groups should be added in Figure 8
Minor comments:
1. Page 4, Line 138 “Micrurus corallinus”; Page 10, Line 353, “M. corallinus” should be italics
2. Page16, Line 633 “constant agitation (200 x g)” or 200 rpm?
3. Some restriction endonuclease. Page 16, Lines 611, 616, 619, and 621 “EcoRI “and “NotI” should be “EcoRI” and “NotI”, respectively.
Author Response
Dear Reviewer.
Thank you for your diligent review of our manuscript. We appreciate the time and effort you have invested in providing your valuable feedback.
Below, we have addressed each of your comments in detail:
1. For cysteine-rich polypeptide toxins, it is theoretically possible to form a variety of isomers. Whether chemically synthesized or recombinant peptides, how can the authors identify the disulfide bridge of the final product?
R: As the reviewer correctly pointed out, the folding process of the three-finger toxin can lead to various isomers due to different disulfide bond pairings.
However, for chemically synthesized toxins, the most thermodynamically stable peptide is preferentially produced and corresponds to the most abundantly purified oxidized toxin. The biological activity of sNXH8 on muscle-type nAChRs validates the correct functional folding of this toxin, as a misfolded toxin would not retain functional properties.
Regarding recombinant toxins, although different studies have successfully obtained correctly folded and biologically active toxins [e.g. Yamauchi, Y et al., 2016. DOI: 10.1080/09168451.2015.1065169, and Nys, M. et al., 2022. DOI: 10.1038/s41467-022-32174-7], achieving this is not straightforward. Prokaryotic cells lack much of the necessary machinery for proper folding of cysteine-rich proteins and are also not conducive to providing an optimal redox environment.
Moreover, the extension in the N-terminus or C-terminus generally impairs correct folding in the recombinant constructs concerning 3FTx. Therefore, it is not surprising that only chemically synthesized toxins have demonstrated antagonistic activity over nAChRs, likely due to incorrect folding or refolding.
Lastly, identifying disulfide bridges in the final product would definitively require resolving the toxin’s three-dimensional structure to confirm the arrangement of the five disulfide bonds.
2. In the Materials and Methods section, more details should be given on peptide synthesis and oxidative folding procedures. For example, Cysteine protecting groups (Acm or Trt), and the step of oxidative folding. The in vitro oxidative folding of disulfide-rich proteins can be challenging.
R: According to the reviewer’s comments, the solid-phase toxin synthesis and refolding processes have been described in greater detail in the Materials and Methods section. The protecting groups used for the different amino acids were specified, such as trityl for cysteine, and the refolding step was detailed.
3. The recombinant 6xHis-NXH8 fusion protein has a theoretical molecular mass of 11,399 Da. However, the SDS-PAGE results alone were not sufficient to verify the correctness of the recombinant protein. HPLC and mass spectrometry are recommended for purification and identification of recombinant proteins.
R: Indeed, SDS-PAGE alone may not conclusively demonstrate that the strongest protein band is NXH8. However, examining the cross-reactivity results of anti-rNXH8 antibodies (Figure 7b) reveals that these antibodies recognize a protein in Micrurus corallinus venom that is positioned almost identically to the recombinant protein observed in Figure 6. Notably, figure 6a shows that the prominent band is only present in the E. coli extract post-induction, and all subsequent downstream processes depicted in Figures 6b to 6e were conducted using the same extract. Additionally, in Figure 7a, the SDS-PAGE profiles of various elapid venoms indicates that low mass proteins, which include three-finger toxins, are positioned nearly at the same site as the recombinant protein. As a result, despite the limitations of SDS-PAGE, we believe our data robustly supports the correctness of the recombinant protein.
4. Although studies have shown that NXH8 interacts with skeletal muscle nAChRs, its specificity, and selectivity for other subtypes of nAChRs were not evaluated.
R: We agree with your observation. However, due to the limited availability of synthetic toxins, primarily attributed to their high acquisition costs, our focus has been directed towards studying skeletal muscle nicotinic receptors.
Indeed, we are currently conducting a comprehensive in vivo study to evaluate the lethality of NXH8 and the effectiveness of anti-sNXH8 antibodies in neutralizing M. corallinus venom. As part of this new study, we also plan to incorporate AI-powered in silico analyses, which will involve advanced molecular modeling of NXH8 and its interactions with various subtypes of nicotinic acetylcholine receptors (nAChRs).
Depending on the outcomes of these in silico studies, we may then proceed to further assess - in vitro - the interactions of sNXH8 with these additional receptor subtypes.
5. Statistical analysis and comparison of different concentration groups should be added in Figure 8.
R: Figure 8 was adjusted as requested. Sample size and statistical data were included. X-axis was revised for a better understanding, as well. Minor comments:
1. Page 4, Line 138 “Micrurus corallinus”; Page 10, Line 353, “M. corallinus” should be italics.
The text have been adjusted.
2. Page16, Line 633 “constant agitation (200 x g)” or 200 rpm?
Thank you for your observation. The correct is 200 rpm. The text was adjusted.
Some restriction endonuclease. Page 16, Lines 611, 616, 619, and 621 “EcoRI “and “NotI” should be “EcoRI” and “NotI”, respectively.
Thank you for your observation. The text was adjusted.